# The Influence of the Protozoan *Giardia lamblia* on the Modulation of the Immune System and Alterations in Host Glucose and Lipid Metabolism

**DOI:** 10.3390/ijms25168627

**Published:** 2024-08-07

**Authors:** Sylwia Klimczak, Kacper Packi, Alicja Rudek, Sylwia Wenclewska, Marcin Kurowski, Daniela Kurczabińska, Agnieszka Śliwińska

**Affiliations:** 1Department of Nucleic Acid Biochemistry, Medical University of Lodz, 92-213 Lodz, Poland; 2AllerGen Center of Personalized Medicine, 97-300 Piotrkow Trybunalski, Poland; 3Diabetology and Internal Medicine Department, Provincial Hospital in Sieradz, 98-200 Sieradz, Poland; 4Department of Immunology and Allergy, Medical University of Lodz, 92-213 Lodz, Poland; 5Preveneo Health Medicine Center, 40-045 Katowice, Poland

**Keywords:** *Giardia lamblia*, giardiasis, metabolic disorders, insulin resistance, inflammation

## Abstract

*Giardia lamblia*, the cause of giardiasis, significantly impacts patients with metabolic disorders related to insulin resistance (IR). Both giardiasis and metabolic disorders share elements such as chronic inflammation and intestinal dysbiosis, which substantially affect the metabolic and cytokine profiles of patients. This review discusses the mechanisms of virulence of *G. lamblia*, its influence on the immune system, and its association with metabolic disorders. The review aims to show how *G. lamblia* invasion acts on the immune system and the glucose and lipid metabolism. Key findings reveal that *G. lamblia* infection, by disrupting intestinal permeability, alters microbiota composition and immune responses, potentially impairing metabolic status. Future research should focus on elucidating the specific mechanisms by which *G. lamblia* influences the metabolism, exploring the long-term consequences of chronic infection, and developing targeted therapeutic strategies that include both parasitic and metabolic aspects. These insights underscore the need for a multidisciplinary approach to the treatment of giardiasis in patients with metabolic disorders.

## 1. Introduction

Giardiasis, caused by the protozoan parasite *Giardia lamblia*, is one of the most common parasitic infections worldwide, affecting millions of people each year. It is responsible for approximately 280 million cases of diarrhea annually, with a significant impact on both developing and developed countries [1]. The prevalence of giardiasis varies significantly by region, with infection rates reaching as high as 30–40% in Sub-Saharan Africa, South Asia, and Latin America, primarily due to poor sanitation and limited access to clean water [2]. In contrast, the prevalence in regions with higher sanitary standards, such as the United States and European countries, is around 2–5% [3,4,5], although the exact number of cases remains uncertain due to underreporting and the prevalence of asymptomatic infections [5].

*G. lamblia* colonizes the small intestine, causing symptoms ranging from asymptomatic carriage to severe diarrhea and malabsorption [6]. Central to its pathogenicity is the parasite’s ability to attach to the intestinal epithelium and impact the host’s immune system, allowing it to evade detection and sustain infection. *G. lamblia* employs several virulence mechanisms, including adhesion to the intestinal epithelium, induction of apoptosis in enterocytes, secretion of proteases, and reduction in nutrient absorption and secretion [7]. The parasite uses structures like a ventral disk and flagella to attach to intestinal cells, leading to the damaging and shortening of the intestinal villi, which impairs nutrient absorption and triggers inflammatory responses [7].

The immune response to *G. lamblia* infection is characterized by a complex interplay of inflammatory and anti-inflammatory processes. Upon infection, the host’s immune system activates various pathways to control the parasite, including the production of pro-inflammatory cytokines and chemokines. Th17 cells, for instance, play a crucial role in producing IL-17A, a key element in the immune response against *G. lamblia* [8]. However, the parasite employs multiple strategies to modulate the host’s immune response, enabling it to evade detection and persist in the host [9]. Beyond its impact on the immune system, *G. lamblia* infection is also associated with metabolic disorders. Emerging evidence suggests that the parasite may affect carbohydrate and lipid balance, potentially contributing to exacerbating metabolic abnormalities in conditions such as obesity, insulin resistance, and type 2 diabetes [10,11,12]. For instance, infection with *G. lamblia* has been shown to activate the AKT pathway, affecting glucose and insulin levels in the host [13]. Additionally, the parasite’s disruption of gut microbiota balance during infection can lead to metabolic disturbances, highlighting the interconnectedness of giardiasis and metabolic health [9,14].

This review discusses the mechanisms of virulence of *G. lamblia*, its influence on the immune system, and its association with metabolic disorders. By analyzing the pathophysiological mechanisms and potential consequences of *G. lamblia* infection for patients with metabolic disorders, we aim to provide a comprehensive understanding of the complex clinical presentation of giardiasis. This review incorporates findings from cell line experiments, animal models, and human studies, focusing on both *G. lamblia* in humans and *G. muris* in mice, covering research published from 2018 to 2024. Through this review, we aim to elucidate the multifaceted interactions between *G. lamblia* and the host’s immune and metabolic systems, contributing to a better understanding of giardiasis and informing future research and therapeutic strategies.

## 2. Biology and Morphological Features of *Giardia lamblia*


*Giardia lamblia* is a cosmopolitan flagellate belonging to the family *Giardiidae*. Although Antoni van Leeuwenhoek was the first to describe *Giardia* in 1681, the discovery of this protozoan is credited to the Czech physician Vilem Lambl in 1859 [15]. The species was named “lamblia” by Blanchard in 1888 [16].

*G. lamblia* occurs in two developmental forms: vegetative (trophozoite), and resistant (cyst). Trophozoite is the active form of the parasite. It has a pear-shaped morphology and typically measures about 10–20 μm in length [17]. Under the microscope, two nuclei are visible inside the trophozoite. In the anterior part, there is an adhesive organ called the ventral disk. The ventral disk, composed of microtubules, allows for the attachment to the surface of the intestinal villi. It also possesses four pairs of flagella (posterior, dorsal, ventral, lateral), facilitating movement in the small intestine [17]. Cysts of *G. lamblia* are extremely resilient developmental forms of the parasite. They have an oval shape and are slightly larger than trophozoites, measuring about 8–12 μm in length [18]. Their thick wall provides protection against adverse external conditions, such as temperature changes or the action of chemical substances [18]. The cyst also contains two or four nuclei visible under the microscope and, similarly to trophozoites, adhesive organs on the exterior. Both developmental forms of the parasite play a significant role in its life cycle and transmission.

*G. lamblia* exhibits eight unique genotypes (usually designated as assemblage A–H) [19,20]. Each genotype can cause symptoms of giardiasis, although some are pathogenic. Genotypes A and B are the most common in humans and are found worldwide [21]. Genotypes C–H are less common, and their prevalence depends on the geographical region [19,22]. The genotypes C–H are characteristic to animal hosts, and are primarily associated with infections in animals such as dogs, cats, cows, sheep, pigs, and others [22]. Some of them can also be transmitted to humans, for example, through contact with the feces of infected animals or contaminated water. This results in an overlap of human and animal genotypes, which means that infection with the same species of parasite can occur in both humans and animals.

*G. lamblia* has a developmental cycle that occurs within a single organism (Figure 1). The minimum inoculum dose required to infect a human range from 10 to 100 cysts and cysts can survive in the environment for up to 3 months in water at 4 °C [16,23]. One gram of feces can contain 20,000–150,000 cysts of the protozoan [16].

## 3. Virulence Mechanisms of *Giardia lamblia*

*G. lamblia* employs a sophisticated array of virulence factors to establish infection and cause disease in the host intestine. Understanding these mechanisms is crucial for comprehending the pathogenesis of giardiasis and its effects on human health. The virulence factors of *G. lamblia* play significant roles in host–parasite interactions and profoundly impact the gut microbiota, leading to a range of intestinal and extra-intestinal symptoms. The parasite’s ability to adhere to the intestinal lining, induce cell death, secrete proteases, reduce nutrient absorption, and alter gut motility and microbiota composition are pivotal in its pathogenicity [25,26]. These mechanisms collectively contribute to the complex clinical presentation of giardiasis, which often mimics chronic gastrointestinal disorders such as irritable bowel syndrome, chronic food allergies, and celiac disease [27,28,29,30].

### 3.1. Adhesion and Colonization

The initial step in *G. lamblia* infection involves the parasite’s adhesion and colonization of the intestinal epithelium. The parasite utilizes several structures and proteins for adhesion and colonization of the intestinal epithelium. Physical contact between trophozoites and enterocytes leads to the damaging and shortening of the intestinal villi, which can affect absorption ability and trigger inflammatory responses [31]. *G. lamblia* employs adhesive structures like a ventral disk and flagella to attach to intestinal epithelial cells. The ventral disk functions like a suction cup, while the flagella facilitate movement and strong adherence to the microvilli. Adhesion proteins such as giardins, lectins, and variant surface proteins (VSPs) form chemical bonds with intestinal epithelial cells [25]. VSPs, cysteine-rich proteins covering the parasite’s surface, can be rapidly switched to evade host IgA-directed clearance [14]. 

### 3.2. Induction of Apoptosis

*G. lamblia* can disrupt the cohesion of intestinal epithelial cells through the process of enterocyte apoptosis. The parasite activates caspase-3, -8, and caspase-9, and the pro-apoptotic Bcl-2-associated X protein (BAX), while simultaneously reducing the expression of the anti-apoptotic protein Bcl-2 [32,33]. Poly ADP-ribose polymerase (PARP), a key participant in programmed cell death, contributes to *Giardia*-induced host–cell apoptosis, while another part is not [34]. One of the barrier parameters that can be regulated in an apoptosis-dependent manner is the disruption of cellular ZO-1 [35]. ZO-1 plays a scaffolding protein role. However, the villi cleavage induced by a protease similar in structure to cathepsin by Giardia is independent of caspase-3 activity [35]. Therefore, in giardiasis, part of the epithelial pathophysiology is dependent on apoptosis, while part of it is not. Activation of enterocyte apoptosis and impairment of epithelial barrier function by *Giardia* may be associated with specific parasite strains [36]. After parasite eradication in a murine model of giardiasis, small intestinal permeability returns to baseline [36]. An intriguing aspect is that sodium and glucose transport proteins, such as SGLT-1, play a protective role in the process of enterocyte apoptosis during infestation. In fact, by enhancing the activity of epithelial SGLT-1 and thus increasing glucose uptake, enterocytes attempt to protect against apoptosis induced by *Giardia* [37].

### 3.3. Proteases and Metabolic Enzymes

*G. lamblia* secretes various proteases, including cathepsin B-like cysteine proteases and giardipain-1, which degrade host proteins, disrupt the epithelial barrier, induce apoptosis in intestinal epithelial cells, and modulate host immune responses [25,38]. The parasite employs specific metabolic enzymes to compete for nutrients and modulate host responses. Arginine deiminase (ADI) and ornithine carbamoyltransferase (OCT) deplete arginine, potentially impairing intestinal epithelial cell proliferation and nitric oxide production [25,38]. 

### 3.4. Disruption of Absorption and Secretion

Studies conducted on in vivo and in vitro models have shown that *G. lamblia* causes disturbances in the absorption of glucose, sodium, and water, and reduces disaccharidase activity by reducing the absorptive surface area of the epithelium [39]. Moreover, parasite metabolites can penetrate the epithelial barrier, causing brush border damage and disaccharidases deficiencies, leading to diarrhea. Recent research suggests that *Giardia* may affect chloride secretion responses in human colonic cells in vitro and in mouse models [40]. 

### 3.5. Impact on Gut Motility and Microbiota

*G. lamblia* affects intestinal motility by limiting the movements of intestinal villi and muscle contractions, disrupting the coordination of villi movements and disturbing the balance of gut microbiota, which becomes increasingly evident in the context of the microbiota’s role in controlling *G. lamblia* invasion [41,42,43]. Trophozoites compete with the commensal microbiome for nutrients and ecological niches in the duodenal microenvironment, with resident bacteria playing a role in *G. lamblia* colonization [14]. New research indicates that *Giardia* disrupt gut flora balance during the acute phase of infestation, affecting homeostasis in the large intestine [9,41,42,43]. The parasite-induced intestinal invasion alters the composition of the microbiota, summarized in Table 1. Studies show that samples positive for *Giardia* correlate with dysbiosis states, including an increase in harmful bacterial species like *Escherichia coli* and *Enterococcus* spp., and a link between enterobacterial colonization and severe malabsorption syndrome in symptomatic giardiasis [44,45]. *Giardia* infections result in several changes to the intestinal microbiota, such as disruption of microbial biofilm structure, altered virulence in commensal species, and changes in species abundance and diversity, which influence *Giardia* pathogenesis by affecting colonization resistance, immune responses, and brush border defects. Specific bacterial taxonomic shifts associated with *Giardia* infections include increased abundance of bifidobacteria and enterobacteria (e.g., *Klebsiella pneumoniae*, *Enterobacter cloacae*, *and Enterobacter hafniae*) in patients with severe intestinal malabsorption, lower microbial diversity and species richness, a shift from enterotype I (*Bacteroides-dominated*) to enterotype II (*Prevotella-dominated*), and decreased abundance of microbiome Vitamin B12 biosynthesis genes [46,47]. Multiple studies show an increase in *Prevotella* abundance associated with *Giardia* infection, which appears to be a robust signature of *Giardia* colonization, and is linked to the promotion of Th17 cell differentiation, crucial for controlling *Giardia* infection [48,49]. Other notable microbiome changes during *Giardia* infection include reductions in *Gammaproteobacteria*, *Natronobacillus*, *Lactobacillus*, and *Leuconostoc*, and increases in *Lachnospiraceae*, *Ruminococcus*, and *Clostridiales*, highlighting the complex interactions between the parasite, host, and commensal bacteria [50,51].

The combined effects of *Giardia’s* virulence factors on intestinal motility and microbiota composition interact to induce intestinal inflammation and disturb the absorption of lipids, carbohydrates, and other essential nutrients. These disturbances can manifest as immune reactions and metabolic disorders, including decreased glucose tolerance and reduced insulin production [40]. *G. lamblia* may disrupt the absorption of vitamin B12 by damaging intestinal villi or competing for nutrients, potentially leading to a deficiency of this vitamin. In many reports, giardiasis is indicated as a potential cause of vitamin B12 deficiency [52,53]. Additionally, infected individuals experience a decrease in serum levels of copper, zinc, and magnesium [54,55]. The results of numerous studies on the levels of trace elements in giardiasis have shown a significant decrease in zinc concentration and a simultaneous increase in copper concentration [54,55]. Some researchers point to possible iron absorption disorders as a complication of *G. lamblia* infestation. Vitamin A deficiency is multifactorial, but giardiasis has been significantly associated with this deficiency [56,57]. Additionally, giardiasis may interfere with the absorption of folic acid [58].

The overall impact of these mechanisms contributes to the complex clinical presentation of giardiasis, including both gastrointestinal symptoms as potential systemic effects.

**Table 1 ijms-25-08627-t001:** Changes in gut microbiota composition induced by *G. lamblia* infection.

Bacterial Species/Strains	Function	Change (+/−)	Relation to Microbiota Biodiversity	Ref.
** *Escherichia coli* **	Pathogenic, can cause intestinal infections	**+**	Increase in harmful bacteria, contributing to dysbiosis and reduced gut health.	[59,60]
***Enterococcus* spp.**	Opportunistic pathogens, can cause infections	**+**	Increase in harmful bacteria, leading to severe malabsorption syndrome and dysbiosis.	[59,60]
** *Bifidobacteria* **	Beneficial, involved in digestion and immunity	**+**	Altered balance with increased bifidobacteria can affect gut homeostasis.	[47,61]
** *Enterobacteriaceae* **	Mixed, some pathogenic species	**+**	Increase in enterobacteria like *Klebsiella pneumoniae*, *Enterobacter cloacae*, and *Enterobacter hafniae* associated with severe malabsorption.	[47,61]
** *Prevotella* **	Beneficial, involved in carbohydrate metabolism	**+**	Increased abundance is a robust signature of *Giardia* colonization and is linked to immune responses like Th17 cell differentiation.	[9,62]
** *Gammaproteobacteria* **	Diverse, includes many pathogens	**−**	Reduction in *Gammaproteobacteria* affects the overall balance and health of the microbiota.	[9,26]
** *Natronobacillus* **	Not well-defined	**−**	Decrease affects overall microbial diversity.	[9,26]
** *Lactobacillus* **	Beneficial, promotes gut health	**−**	Decrease reduces beneficial bacteria, affecting gut health and balance.	[9,26]
** *Leuconostoc* **	Beneficial, involved in fermentation	**−**	Decrease reduces beneficial bacteria, affecting gut health and balance.	[9,26]
** *Lachnospiraceae* **	Beneficial, involved in short-chain fatty acid production	**+**	Increase may compensate for some loss of beneficial bacteria, but overall contributes to a complex interaction.	[9,26]
** *Ruminococcus* **	Beneficial, involved in fiber digestion	**+**	Increase contributes to altered gut flora balance.	[9,26]
** *Clostridiales* **	Mixed, includes both beneficial and harmful species	**+**	Increase highlights complex interactions and potential shifts towards dysbiosis.	[9,26]

“−” indicates a decrease in the number of a particular bacterial species/strain due to *G. lamblia* invasion; “+” indicates an increase in the number of a particular bacterial species/strain due *to G. lamblia* invasion.

## 4. Host Inflammatory Response and Its Modulation

*G. lamblia* induces an inflammatory response in the host through mechanical damage to enterocytes and the release of toxic compounds, leading to a range of immune reactions [63]. Upon colonization of the small intestine, *G. lamblia* attaches to the epithelial cells, causing physical disruption and initiating an immune response. Significant advances in recent years have improved our understanding of the mechanisms that initiate the innate immune response to this protozoan infection and how this response contributes to the development of adaptive immunity. *G. lamblia* infection triggers a robust adaptive response, involving the production of specific IgA antibodies and the activation of CD4+ T cells [64]. Recent studies have revealed the crucial role of interleukin 17 (IL-17) in coordinating the immune response and the importance of gut microbiota in the disease process [65]. Therefore, *G. lamblia* employs multiple strategies to modulate the host’s immune system, facilitating its survival and persistence within the host. This chapter explores the mechanisms through which *G. lamblia* induces and modulates inflammatory responses, the role of different immune cells and cytokines, the implications for the pathogenesis of giardiasis, and the influence of environmental factors on the host immune response.

### 4.1. Innate Response

The innate response against *G. lamblia* represents the body’s first line of defense against this parasite. When *G. lamblia* trophozoites enter the small intestine, they are quickly recognized by various components of the innate immune system, including immune cells such as macrophages, dendritic cells (DCs), neutrophils, and natural killer (NK) cells. These cells utilize pattern recognition receptors (PRRs), particularly Toll-like receptors (TLRs) such as TLR2 and TLR4, to detect the parasite’s characteristic molecular patterns (PAMPs) [66]. This recognition initiates a cascade of defensive reactions, activating relevant signaling pathways in the cells. These pathways lead to the production of pro-inflammatory cytokines like IL-12, TNF-α, IL-6, and interferons, as well as chemokines and other inflammatory mediators [67]. Additionally, the complement system is activated, contributing to parasite opsonization and lysis [68]. Antimicrobial peptides, such as defensins and cathelicidins, are released by epithelial cells and immune cells, providing direct antimicrobial activity against *G. lamblia* [69]. The innate immune response also involves the activation of the inflammasome, a multiprotein complex that further enhances the inflammatory response and promotes the secretion of IL-1β and IL-18 [70]. These coordinated actions of the innate immune system not only provide immediate defense against the parasite, but also help to bridge the gap between innate and adaptive immunity by activating and shaping the subsequent adaptive immune response. The recombinant arginine deiminase protein (ADI), secreted during host–parasite interactions, also activates TLR2 and TLR4 in reporter HEK293 cell lines [71]. Another parasite protein, BiP, has been identified as a TLR4 ligand [72]. 

Parallel to TLR activation, NLRP3 inflammasomes in immune cells are activated. Macrophages, particularly peritoneal macrophages and bone marrow-derived macrophages, show significant NLRP3 inflammasome activation when exposed to *G. lamblia* trophozoites or their secreted components [38,70]. This includes activation in the murine macrophage cell line J774A.1, as well as in primary mouse peritoneal macrophages [38,70]. Monocytes are also capable of NLRP3 inflammasome activation upon *Giardia* stimulation, although this has been primarily demonstrated with other pathogen-associated molecular patterns rather than specifically with *G. lamblia* [73]. NLRP3 inflammasome complexes play a key role in the production of IL-1β and IL-18, cytokines that significantly enhance the inflammatory response [38,70]. These cytokines recruit additional immune cells such as neutrophils, monocytes, and T cells to the site of *G. lamblia* infection in the intestinal mucosa [38]. This coordinated immune cell recruitment and activation helps to mount an effective defense against the parasite.

The complement system also actively participates in combating *G. lamblia*. The lectin pathway, in particular, is crucial, with mannose-binding lectin (MBL) recognizing and attaching to the surface of trophozoites [74]. This leads to the activation of the complement cascade, which can directly damage *G. lamblia* trophozoites by forming the membrane attack complex (MAC) and attract and activate other immune cells, including mast cells [75]. The innate response against *G. lamblia* is rapid and nonspecific, but effectively limits the parasite’s spread in the early stages of invasion. Simultaneously, through cytokine production and antigen presentation, it sets the stage for a more specific adaptive response essential for the complete elimination of *G. lamblia*.

### 4.2. Adaptive Response

The adaptive response against *G. lamblia* involves both cellular and humoral immune mechanisms. CD4+ T lymphocytes, particularly Th17 and Th1 cells, play a crucial role in coordinating this response. Th17 cells stimulate neutrophil recruitment and prompt intestinal epithelial cells to produce antimicrobial peptides through IL-17 production, while Th1 cells secrete IFN-γ, activating macrophages [65]. Macrophages, in addition to their phagocytic functions, secrete cytokines that influence the inflammatory response. In the context of *G. lamblia* infection, macrophages can produce cytokines such as IL-6, TNF-α, and IL-12 p4 [13]. 

The humoral response includes the production of IgM, IgG, and IgE antibodies, with IgA production being particularly stimulated IgA antibodies appear as early as three days post-infection, peaking between the 6th and 9th days of the disease [76]. These antibodies target the parasite’s surface and intracellular antigens, hindering trophozoite adherence to epithelial cells, causing their agglutination, and inhibiting movement. The nonspecific hypergammaglobulinemia characteristic of parasitic infections results from polyclonal B cell stimulation by parasite antigens acting as mitogens, potentially leading to B cell exhaustion [77]. IgM antibodies are typically the first to be produced in response to *G. lamblia* infection, playing a crucial role in the early stages of the immune response [78]. They are effective in agglutinating the trophozoites and facilitating their removal by the immune system. IgG antibodies are produced later and provide long-term immunity [42]. They are involved in neutralizing the parasite and opsonizing it for phagocytosis by macrophages. IgE antibodies, although less prominent in the response to *G. lamblia*, can contribute to the immune defense by mediating allergic reactions and facilitating the recruitment of other immune cells, such as mast cells, to the site of infection [26].

In turn, CD8+ T lymphocytes have a dual impact: they contribute to parasite elimination, but can also cause tissue damage in the intestine. Other mechanisms, such as the phagocytosis of trophozoites by macrophages, the cytotoxic activity of lymphocytes, and complement activation, also participate in eliminating *G. lamblia* [79]. The interplay of these diverse mechanisms allows for an effective fight against the infection; however, the complexity of the immune response can also contribute to chronic disease forms when the balance between parasite elimination and host tissue damage is disrupted.

### 4.3. Cytokines and Chemokines

Animal models have shown that *G. lamblia* infection stimulates the production of various cytokines, both inflammatory and anti-inflammatory, including IL-2, IL-4, IL-10, IL-13, IL-17, IL-22, TNF-α, and IFN-γ [80] (Table 2). Studies on infected individuals have reported diverse cytokine responses, with some reporting elevated levels of IL-2, IL-4, and IL-10, and others reporting decreased IL-4 levels [81,82]. Th17 cells play a crucial role in the immune response to *G. lamblia* infection by producing IL-17A, which induces the expression of antimicrobial peptides in the mucosa and recruits neutrophils [83]. Studies in mice and humans have confirmed the importance of IL-17A and its receptor IL-17RA in controlling infection and supporting IgA antibody responses [64]. In mice infected with the parasite, an increase in IL-17 mRNA levels in the small intestine was observed three weeks post-infection, along with an almost complete absence of IgA in stool samples [8,64]. A study of adult travelers returning to Denmark with giardiasis showed elevated IL-17 responses in vitro [84]. An ineffective Th17 response can lead to prolonged infection, while an overly intense response can cause acute inflammation. A balanced Th17 response is essential for effective parasite clearance, and improper regulation may explain symptom variability and difficulties in elimination [64]. Studies in IL-17RA receptor-deficient mice infected with *G. muris* showed reduced expression levels of genes encoding antimicrobial peptides and mannose-binding lectin, resulting in a defect in parasite elimination [64]. IL-6, produced by DC, supports Th17 cell differentiation, which is crucial for infection control. IL-6-deficient mice have difficulties combating parasites, suggesting that IL-6 promotes the differentiation of IL-17-producing Th17 cells [85,86]. IL-10, produced by various immune cells, acts as a strong anti-inflammatory cytokine, inhibiting macrophage-induced parasite elimination by IFN-γ while protecting host tissues from immune response-related damage [87]. IL-10-deficient mice show more severe infection symptoms, such as diarrhea and colitis, suggesting that IL-10 has a regulatory role in controlling the inflammatory response during giardiasis [30]. During *G. muris* infection in IL-10-deficient mice, an increase in CD11b+ and CD11c- macrophages in the colon was observed, causing colitis, whereas wild-type mice did not show such inflammation [30].

### 4.4. Modulation of Immune Response and Tolerance

*G. lamblia* exhibits a unique ability to modify its surface antigens. This process involves the expression of one of a family of related proteins, called variant specific surface proteins (VSPs), which are periodically exchanged for another [88]. Antigenic variation provides an effective defense mechanism, allowing the parasite to avoid recognition by the host immune system. As a result, *G. lamblia* can successfully survive in the host’s body for an extended period of time, hindering its elimination by immune mechanisms. *G. lamblia* employs various strategies to modulate the host’s immune response, enabling it to successfully colonize the small intestine. The parasite attenuates the inflammatory response through several mechanisms. It consumes L-arginine, a precursor of nitric oxide (NO), an important antiparasitic compound, thereby reducing NO levels and weakening the antiparasitic activity of the mucosa [89]. *G. lamblia* also degrades cytokines and chemokines, including CXCL1, CXCL2, CXCL3, IL-8, CCL2, and CCL20, via cysteine proteases [90,91]. This degradation impedes neutrophil chemotaxis and attenuates the overall inflammatory response [92]. *G. lamblia* trophozoites produce cathepsin B proteases that have the ability to degrade CXCL8, which further inhibits the attraction of neutrophils to sites of infection [92]. Additionally, *G. lamblia* interacts with DC, altering MHC class II expression and reducing their ability to present antigens. It also affects T cells, reducing the production of pro-inflammatory cytokines such as IFN-γ and IL-2 [93]. The parasite can promote Th2 responses over Th1 responses, potentially inducing immune tolerance. Th1 cytokines, such as IL-2, are crucial for parasite elimination, while Th2 cytokines, such as IL-4, have anti-inflammatory properties and stimulate various immune cells, such as eosinophils, basophils, and IgG immunoglobulin production [31,94]. This leads to increased levels of regulatory cytokines such as transforming growth factor (TGF) and IL-10, which help fight parasitic infections [95,96,97]. These immunomodulatory tactics contribute to the survival of the parasite, the chronicity of the infestation, and affect the severity of giardiasis symptoms. By weakening the host immune response, *G. lamblia* creates an environment more conducive to its survival in the gastrointestinal tract.

#### 4.4.1. Effects on the Gut Microbiome

*G. lamblia* infection can lead to tissue damage and an inflammatory response by imbalancing the intestinal microbiota [98]. Studies have shown that in the presence of *G. lamblia*, some bacteria, including *Escherichia coli*, can transform into forms that are toxic to other organisms [99]. In addition, *G. lamblia* causes dysbiosis in mucosal biofilms and increases virulence of commensal microorganisms against human epithelial cells in vitro [50]. An increase in inflammatory markers was observed in mice exposed to *G. lamblia* metabolites, although this effect did not occur with exposure to the live parasite. *G. lamblia* cysteine proteases play a key role in increasing intestinal permeability and decomposing the bacterial biofilm.

On the other hand, the intestinal microbiota affects the ability of *G. lamblia* to colonize the host. Studies have shown that probiotics, such as *Lactobacillus johnsonii* and *L. rhamnosus*, can reduce the growth of *G. lamblia* trophozoites, reduce mucosal damage and parasite burden, and stimulate IgA antibody production [89]. These effects have been observed in various animal models, including mice [100]. Manipulation of the microbiome may be a promising strategy in alleviating the clinical effects of giardiasis.

#### 4.4.2. Influence of Environmental Factors on Host Immune Response

The immune profiles of patients with giardiasis are significantly affected by various environmental factors. Geographic location and socioeconomic status play crucial roles; areas with poor sanitation and limited access to clean water see higher giardiasis prevalence [101]. In endemic areas, continuous exposure to the parasite can lead to a more regulated immune response, characterized by a balance between pro-inflammatory and regulatory cytokines, which may reduce the severity of symptoms [102]. Malnutrition weakens the immune system, reducing essential immune cells like T lymphocytes and impairing antibody production, while overnutrition and obesity result in chronic low-grade inflammation, elevating inflammatory markers such as C-reactive protein (CRP) and interleukin-6 (IL-6) [103]. Microbial co-infections modulate the immune response by either intensifying immune activation against *Giardia* or diverting immune resources, resulting in a mixed profile of suppression and hyperactivation [98].

Environmental toxins and pollutants disrupt normal immune functions, leading to dysregulated cytokine production and impaired immune cell function. For example, heavy metals like lead or mercury can suppress macrophage and lymphocyte activity, prolonging infections, and worsening outcomes [104]. Lifestyle factors, including chronic stress, physical activity, and hygiene practices, further influence immune profiles [105]. Chronic stress elevates cortisol levels, reducing the activity of natural killer (NK) cells and T cells, while regular physical activity enhances immune function by boosting immune cell circulation and increasing anti-inflammatory cytokines [106]. Poor hygiene practices increase the risk of infection, promoting repeated exposures and persistent infections, complicating the immune response [106].

The impact of the *Giardia* parasite on the immune system remains a subject requiring further investigation. The interaction between cytokines and immune cells demonstrates the complex modulation of the host immune system by *G. lamblia* invasion, as illustrated in Figure 2. There is a need for a more precise understanding of changes in the levels of various interleukins during infestation and post-treatment. These cytokines may serve as potential biomarkers, aiding in better monitoring of disease progression and therapy effectiveness. Additionally, it is essential to examine the parasite’s influence on inflammation and metabolic pathway regulation. These changes may have significant consequences for metabolism regulation, leading to long-term health effects. Further research in this area can assist in identifying pathophysiological mechanisms and gaining a better understanding of the interaction between *Giardia* and the host. 

*G. lamblia* adheres to the epithelial cells of the small intestine, causing damage and triggering an inflammatory response characterized by cytokine and chemokine production. The parasite’s pathogen-associated molecular patterns (PAMPs) are recognized by Toll-like receptors (TLRs), particularly TLR2 and TLR4, on host cells. Upon binding to PAMPs, TLR receptors activate signaling pathways that lead to the production of pro-inflammatory cytokines such as IL-1β, TNF-α, and IL-6. G. lamblia modulates this response by consuming L-arginine, degrading cytokines and chemokines with cysteine proteases, and producing proteins like cathepsin B. These actions reduce nitric oxide (NO) levels, hinder neutrophil chemotaxis, and decrease dendritic cell antigen-presenting capacity. Additionally, *G. lamblia* promotes Th2 over Th1 responses, particularly in cases of chronic infection, which is an exception among protozoa. This shift leads to higher levels of regulatory cytokines like IL-10, which inhibit macrophage activity and protect host tissues from excessive inflammation, facilitating chronic infection and persistence. This unique ability to modulate the immune response allows the parasite to effectively evade elimination by the host’s immune system.

## 5. Associations between Giardiasis and Metabolic Disorders

The gut microbiota plays a crucial role in metabolic disorders like obesity, dyslipidemia, dysglycemia, and type 2 diabetes. This section explores the connections between gut dysbiosis, inflammation, and metabolic disorders, including the potential role of *G. lamblia* in these processes. Metabolic disorders result from an imbalance between energy intake and expenditure, leading to the dysregulation of lipid and glucose metabolism. Low-grade inflammation, primarily originating from visceral adipose tissue, is a key factor in these conditions [107]. Excessive fat accumulation, hypertrophy, and dysfunction of adipocytes contribute to the activation of inflammatory cells, which secrete pro-inflammatory cytokines, including interleukins (IL-6, IL-1β, IL-18), tumor necrosis factor-alpha (TNF-α), interferons (IFN-γ), transforming growth factor-beta (TGF-β), and monocyte chemoattractant protein-1 (MCP-1) [107]. These cytokines enhance inflammation in adipose tissue and affect various organs and tissues, contributing to insulin resistance (IR) and other metabolic complications. The chronic low-grade inflammatory state leads to lipotoxicity, systemic inflammation, and metabolic changes, creating a vicious cycle that ultimately results in metabolic diseases.

The gut microbiota is involved in carbohydrate and lipid metabolism, synthesis of vitamins and amino acids, proliferation of epithelial cells, defense against infections, and hormonal regulation. Gut dysbiosis is closely associated with metabolic disorders related to obesity, l and type 2 diabetes [108,109]. Changes in gut microbiota diversity affect lipid metabolism and disrupt gut–brain communication, leading to IR. Gut dysbiosis promotes low-grade inflammation through various mechanisms, contributing to generalized inflammation in metabolic disorders. Individuals with these disorders often exhibit decreased bacterial diversity, increased Gram-negative bacteria, and alterations in specific bacterial strains [110]. This abnormal gut microbiota can increase endotoxin production, enhance gut permeability, and activate the immune system, exacerbating inflammation in adipose tissue and other organs [110]. Obese individuals and those with IR and dyslipidemia typically have less diverse gut microbiota, with decreased proportion of *Bacteroidetes* and a higher level of *Firmicutes* [111]. Specific bacterial species are associated with IR, suggesting a direct role of gut microbiota in the development and perpetuation of metabolic disorders.

Research indicates that intestinal parasites can induce metabolic changes, including dysregulation of glucose and insulin metabolism. Infection with the *G. lamblia* parasite can activate the AKT pathway, affecting glucose and insulin levels in the host’s body [112]. The exact mechanism of this activation is not fully understood, but may involve signaling factors, manipulation of the insulin receptor, or interactions with other host cells [38,113]. Activation of the AKT pathway by the parasite may lead to increased glucose uptake by host cells or changes in glucose and insulin metabolism (Figure 3).

Hyperglycemia and dyslipidemia stimulate pro-inflammatory mechanisms, leading to IR and impaired insulin secretion, increasing the risk of developing type 2 diabetes (T2D) and weakening immune response against parasites [114,115]. Eleuza Rodrigues Machado and her colleagues decided to assess the prevalence of parasites, including *G. lamblia*, and compare their presence in individuals with type 1 and type 2 diabetes [10]. The high prevalence of *G. lamblia* in cases of T2D underscores the importance of regular parasitological examinations in these patients. In a study conducted by Gözde Derviş Hakim, the prevalence of giardiasis was found to be 7% in patients with dyspeptic symptoms and 15% in patients with diabetes [116]. No positive results were observed in the test for the presence of *Giardia* antigen, providing significant evidence of a positive association between indigestion and giardiasis [116]. The results of another study indicated a significant increase in blood glucose, cholesterol, and melatonin levels in men with *G. lamblia* infestation [117]. Diabetic patients are more susceptible to giardiasis, possibly due to weakened immune defenses resulting from gut dysbiosis and inflammation. Additionally, typical giardiasis symptoms may be masked by dysglycemia and dyslipidemia symptoms in diabetic or obese patients.

Melatonin, produced primarily by the pineal gland and other organs, is elevated in patients with giardiasis, correlating with leukocytosis caused by *G. lamblia* [117]. Ghrelin, a hormone regulating hunger and glucose metabolism, is significantly lower in giardiasis patients [117]. This may result from elevated glucose levels. Cholesterol levels are also increased in patients with giardiasis [117]. The relationships between ghrelin and HDL levels and increased lipid peroxidation in patients with *G. lamblia* infestation requires further research. Cholecystokinin (CCK), a peptide hormone stimulating bile and pancreatic juice secretion, shows increased levels in the colon of *Giardia*-infected mice, leading to smooth muscle contractions [118]. Some *Giardia* patients exhibit similar phenomena, suggesting the parasite may use CCK for growth, utilizing bile as a food source [119].

*G. lamblia* may affect lipid metabolism in overweight and obese individuals, although existing data are inconsistent. The parasite can reduce intestinal fat absorption and digestion by damaging microvilli. Some studies report elevated levels of total cholesterol, triglycerides, and fatty acids in *G. lamblia* invasion [12,120], while others indicate reduced lipid parameters [121,122]. *Giardia* is believed to acquire cholesterol from the upper small intestine, suggesting the presence of lipids/cholesterol breakdown factors. Proteomic analysis reveals increased levels of lipid metabolism enzymes in *Giardia* cysts, indicating active lipid metabolism in all developmental stages [123]. This may be necessary for effective excystation into infectious trophozoites.

Increasing evidence suggests that giardiasis may significantly impact carbohydrate, lipid, and hormonal metabolism in the body. These disruptions often correlate with obesity, insulin resistance, and type 2 diabetes, suggesting that *G. lamblia* may affect their underlying symptoms, such as exacerbating lipid dysregulation. Previously, *Giardia* infection was believed to be limited to the intestines and not affect other systems in the body. However, an increasing body of research suggests that the parasite may affect various organs and tissues, leading to alterations in lipid, glucose, and hormone metabolism. Further studies on the mechanisms of *Giardia’s* influence on metabolism and immune response may contribute to a better understanding of the pathogenesis of obesity, insulin resistance, and type 2 diabetes, as well as the identification of new therapeutic targets for these disorders.

## 6. Summary

The picture of immune system disorders caused by giardiasis may vary depending on the patient’s age and concurrent diseases (Figure 4). This could be a reason for delayed diagnosis of giardiasis among individuals with metabolic disorders such as obesity or diabetes. The parasite *G. lamblia* attaches to the intestinal wall, causing tissue damage. This activates the host’s immune system, releasing inflammatory factors and reducing the absorption process of nutrients. Chronic infection can affect the population of immune cells, limiting the population of T lymphocytes and favoring another type of immune response. This can lead to a weakening of the inflammatory response and the induction of a state conducive to immunological tolerance. As a result, *G. lamblia* can survive in the body. Undoubtedly, *G. lamblia* invasion affects two key phenomena associated with metabolic disorders, the inflammatory process and gut dysbiosis. However, due to the complexity of mechanisms involving metabolic changes and inflammation, the impact of *G. lamblia* on these processes in patients suffering from metabolic disorders requires in-depth research. These insights underscore the need for a multidisciplinary approach to the treatment of giardiasis in patients with metabolic disorders.

## Figures and Tables

**Figure 1 ijms-25-08627-f001:**
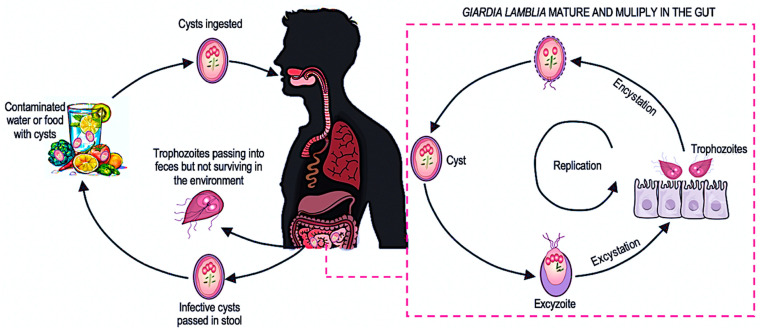
Life cycle of protozoan *Giardia lamblia*. After accidental ingestion, *G. lamblia* cysts reach the stomach, where they undergo a dynamic transformation called excystation under the influence of hydrochloric acid. This process produces a short-lived stage known as an excyzoite, which then differentiates into two vegetative trophozoites. These trophozoites inhabit the mucous membranes of the small intestine, adhering to them with adhesive apparatuses. They reproduce asexually by longitudinal division, forming new colonies in a process known as multiplication. The massive proliferation of trophozoites reduces the absorptive surface of the intestine and damages the intestinal villi, hindering the absorption of nutrients. Under the action of bile acids, trophozoites undergo encystation, transforming back into cysts that are excreted in the feces. In cases of diarrhea, trophozoites that have not yet encysted are also excreted, contributing to the spread of the infection [24].

**Figure 2 ijms-25-08627-f002:**
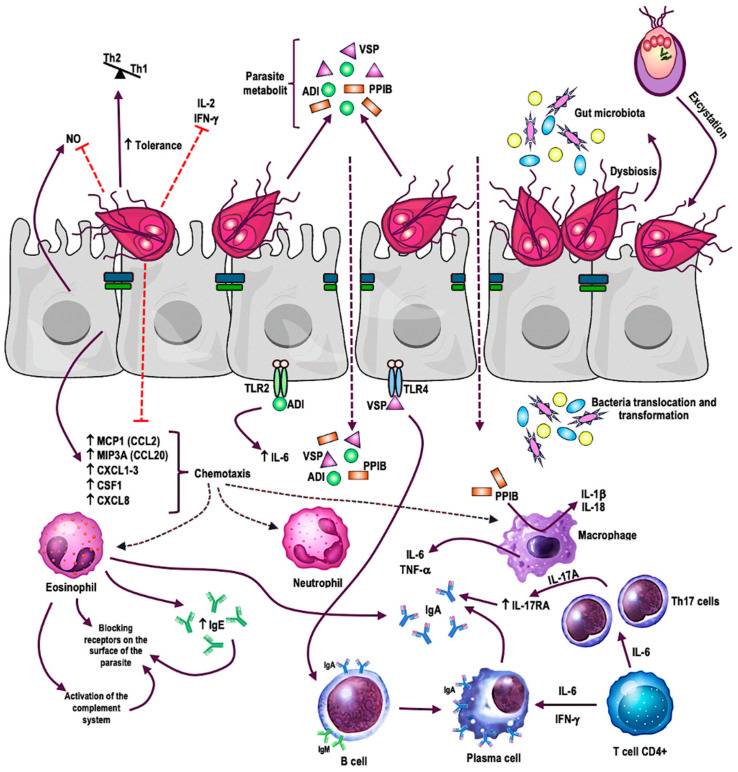
Potential modulation of the host inflammatory response by the protozoan *Giardia lamblia* [25]. Red straight arrows indicate the inhibition; black arrows with arrowheads indicate the direction of influence.

**Figure 3 ijms-25-08627-f003:**
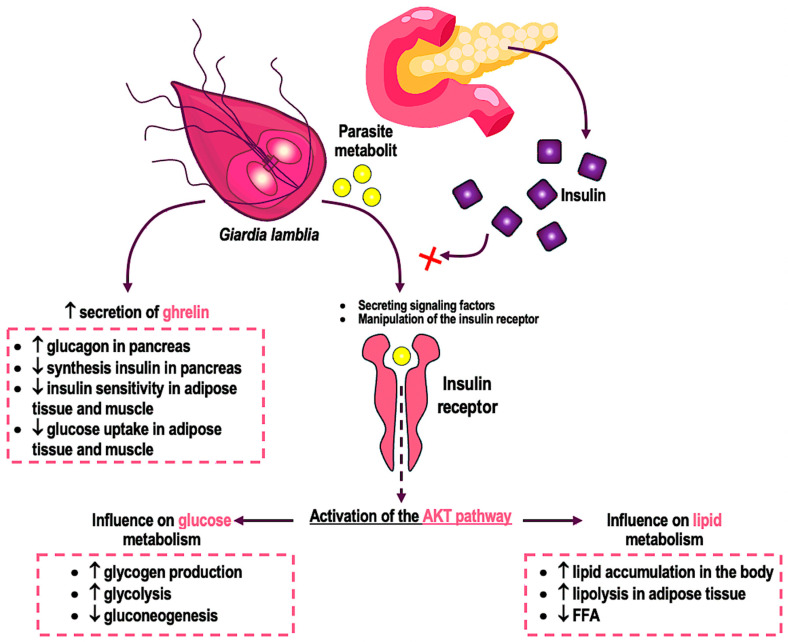
Potential impact of *G. lamblia* metabolites on the regulation of glucose and lipid metabolism. The red cross indicates the inhibition of insulin action by parasite metabolites.

**Figure 4 ijms-25-08627-f004:**
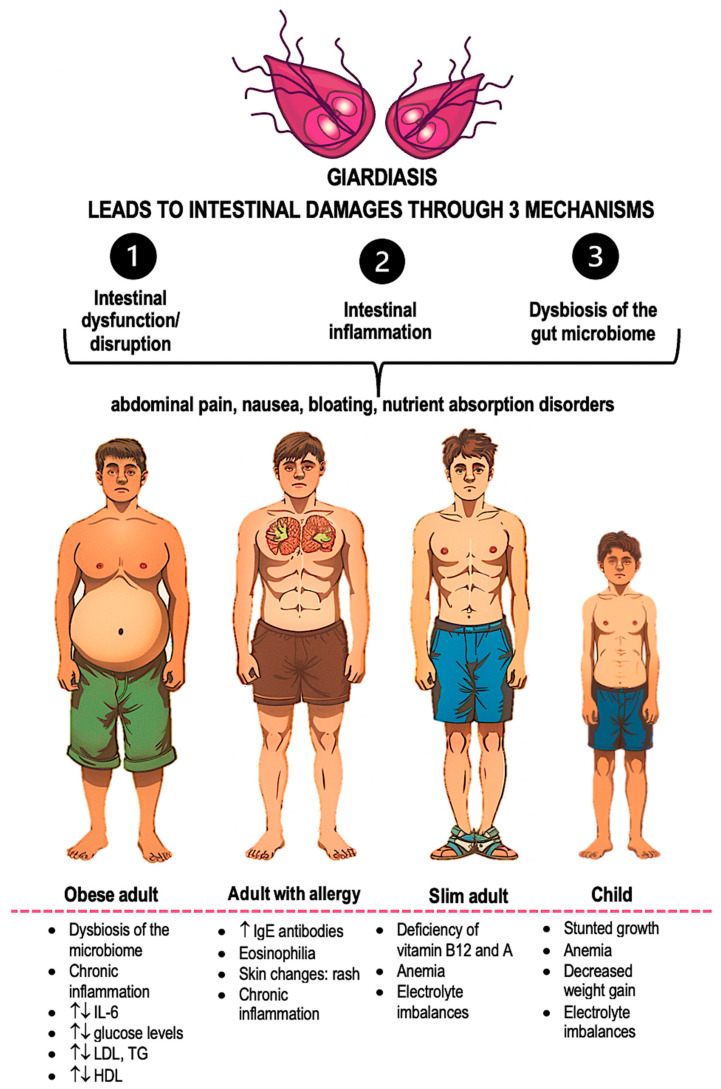
Summary of intestinal damage mechanisms resulting from giardiasis, and symptoms occurring during the course of this disease depending on age and concurrent conditions [61]. An upward arrow indicates an increase; an upward and downward arrow indicates that the change is unknown.

**Table 2 ijms-25-08627-t002:** The inflammatory and anti-inflammatory factors produced by the immune system in response to *G. lamblia* infection.

Inflammatory Factors	Anti-Inflammatory Factors
Elevated levels typically observed.	Concentrations can vary, depending on the stage of infection and the host’s immune response.
IL-2 IL-6 IL-12 p40 IL-17A TNF-α IFN-γ IL-1β IL-8	Prostaglandins Nitric oxide (NO) IL-10 TGF-β

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
