# Peer review of "The Influence of the Protozoan Giardia lamblia on the Modulation of the Immune System and Alterations in Host Glucose and Lipid Metabolism"

_ijms, 2024, doi:10.3390/ijms25168627_

Round 1

Reviewer 1 Report

Comments and Suggestions for Authors

The manuscript titled "The Influence of the Protozoan Giardia lamblia on Modulation of the Immune System and Alterations in Host Glucose and Lipid Metabolism" presents an elegant review on the relationship between the parasite Giardia lamblia, the immune system, and certain metabolic diseases.

The authors provide a current overview of the state-of-the-art regarding the pro- and anti-inflammatory modulation triggered by the parasite when it infects the host. Additionally, they discuss the potential links between obesity, dyslipidemia, type 2 diabetes, and giardiasis. The authors dynamically and coherently showcase the advancements in these research areas while also highlighting limitations that require further investigation.

The figures and tables are excellent.

Despite the high-quality review, one main point needs to be addressed:

The review compiles information on the diagnosis of giardiasis and the interaction of the parasite with the intestinal epithelium. However, the authors do not provide a deeper analysis of these topics in the manuscript, which stands in contrast to the comprehensive discussion on the relationship between the immune system and metabolic disorders. In this context, the influence of the microbiota on giardiasis is mentioned very superficially. It would be more interesting if the authors expanded on topic 7 (Mechanisms of damage to intestinal villi by Giardia lamblia - line 334) to include a more in-depth analysis of the role of the microbiota in giardiasis, incorporating recent information on bacterial modulation, the relationship of the parasite with this niche, and other relevant points. Between lines 630 and 660, the authors describe the importance of the microbiota in general terms without correlating it with giardiasis. Furthermore, removing the entire section on diagnosis from the manuscript would prevent it from diverging from the other topics discussed.

Below are some minor errors that need to be corrected throughout the text:

  1. The entire section 1 (Introduction) lacks references; please add them.
  2. The term "disruption" used to refer to nutritional absorption (line 36) is not appropriate; use alteration or even reduction instead.
  3. Did the authors use any method for excluding articles? Are all the information related to Giardia lamblia infection, or does it include other species? This needs to be clarified in section 2 (Methodology).
  4. In line 66, the correct term is Giardia intestinalis or duodenalis, but it is written as Lamblia intestinalis.
  5. In line 71, correct the name Lambl; the letter "b" is missing.
  6. In line 77, the concept of an organ is used to refer to the ventral disc; it is better to describe it as a cytoskeletal structure.
  7. In line 83, the cyst can contain 2-4 nuclei, not necessarily 2; please correct this.
  8. The paragraph addressing Giardia assemblages (lines 87-98) contains a significant error: there are 8 Giardia assemblages (A-H) present in different host species. A-B is more common in humans but not exclusive to them, which is why giardiasis is considered a zoonosis, as the authors highlight. There are not two groups of genotypes divided between humans and animals. This needs correction.
  9. In lines 111-112, it is stated that two trophozoites are released during the excystation of the parasite, which is incorrect. The figure 1 itself indicates the release of an excyzoite (representing a trophozoite that has not completed cell division, which occurs shortly after excystation). This should be corrected.
  10. The paragraph between lines 154 and 160 needs a reference.
  11. In line 250, explain the abbreviation IFA.
  12. The paragraph between lines 306-321 needs references.
  13. In lines 312-313, it is mentioned that culturing Giardia requires complex equipment. In reality, a simple incubator and the appropriate culture medium are sufficient.
  14. In section 7, page 9, replace "discoid shields" with "ventral disc" for accuracy.
  15. The full name of a species should only be written once in the text (Giardia lamblia); thereafter, the abbreviated form (G. lamblia) should be used. Please correct this throughout the text.

Author Response

Szanowny Recenzencie,

DziÄ™kujemy za przejrzenie naszego manuskryptu zatytuÅ‚owanego „WpÅ‚yw pierwotniaka Giardia lamblia na modulacjÄ™ ukÅ‚adu odpornoÅ›ciowego i zmiany w metabolizmie glukozy i lipidów u gospodarza” i za cenne komentarze, które znaczÄ…co poprawiÅ‚y jakość naszej pracy. Skrupulatnie poprawiliÅ›my nasz manuskrypt, aby zająć siÄ™ problemami, które Pan podkreÅ›liÅ‚. Wszystkie zmiany zostaÅ‚y wprowadzone przy użyciu funkcji „Å›ledzenia zmian” dla PaÅ„stwa wygody. Poniżej znajdÄ… PaÅ„stwo nasze odpowiedzi na komentarze.

Numer rękopisu: IJMS-3076264

Punkt 1:

„PrzeglÄ…d gromadzi informacje na temat diagnostyki giardiozy i interakcji pasożyta z nabÅ‚onkiem jelitowym. Autorzy nie przedstawiajÄ… jednak gÅ‚Ä™bszej analizy tych tematów w manuskrypcie, co stoi w sprzecznoÅ›ci z kompleksowÄ… dyskusjÄ… na temat zwiÄ…zku miÄ™dzy ukÅ‚adem odpornoÅ›ciowym a zaburzeniami metabolicznymi. W tym kontekÅ›cie wpÅ‚yw mikrobioty na giardiozÄ™ jest wspominany bardzo powierzchownie. ByÅ‚oby ciekawiej, gdyby autorzy rozszerzyli temat 7 (Mechanizmy uszkodzenia kosmków jelitowych przez Giardia lamblia - wiersz 334) o bardziej dogÅ‚Ä™bnÄ… analizÄ™ roli mikrobioty w giardiozie, wÅ‚Ä…czajÄ…c najnowsze informacje na temat modulacji bakteryjnej, zwiÄ…zku pasożyta z tÄ… niszÄ… i inne istotne punkty. PomiÄ™dzy wierszami 630 i 660 autorzy opisujÄ… znaczenie mikrobioty w kategoriach ogólnych, nie korelujÄ…c jej z giardiozÄ…. Ponadto usuniÄ™cie caÅ‚ej sekcji dotyczÄ…cej diagnostyki z manuskryptu zapobiegÅ‚oby jej odbieganiu od innych omawianych tematów”.

Odpowiedź: Bardzo doceniamy TwojÄ… dogÅ‚Ä™bnÄ… analizÄ™ naszego manuskryptu i wskazanie bÅ‚Ä™dów, które mogÅ‚yby utrudnić czytelnikom zrozumienie naszej pracy. CaÅ‚kowicie usunÄ™liÅ›my sekcjÄ™ dotyczÄ…cÄ… diagnostyki G. lamblia , co zaowocowaÅ‚o krótszym manuskryptem, który pozwala czytelnikom szybciej skupić siÄ™ na gÅ‚ównym celu przeglÄ…du: powiÄ…zaniu giardiozy ze zmianami w profilach cytokin i zaburzeniami metabolicznymi. RozdziaÅ‚ 7 (obecnie rozdziaÅ‚ 3) dotyczÄ…cy mechanizmów wirulencji G. lamblia zostaÅ‚ rozszerzony o szczegóÅ‚owÄ… analizÄ™ roli mikrobioty w giardiozie, zgodnie z sugestiÄ… Recenzenta.

Punkt 2:

Comment 1: “The entire section 1 (Introduction) lacks references; please add them.”

Response: We kindly thank Reviewer for valuable feedback regarding the Introduction section. References have been added in accordance with the Reviewer's recommendations. Additionally, we have revised the introduction. We appreciate your attention to detail and acknowledge this oversight on our part.

Comment 2: “The term "disruption" used to refer to nutritional absorption (line 36) is not appropriate; use alteration or even reduction instead.”

Response: We thank the Reviewer for this comment. The term "disruption" used in reference to nutrient absorption (line 43 and 125) has been changed according to the Reviewer's suggestion. The change is visible in the revised manuscript.

Comment 3: “Did the authors use any method for excluding articles? Are all the information related to Giardia lamblia infection, or does it include other species? This needs to be clarified in section 2 (Methodology).”

Response: We thank the Reviewer for the suggestion. We have included information about the species in the section describing the methodology used. In the human studies described in the article, the species Giardia lamblia was used. In the study involving mice with induced giardiasis, other researchers utilized the species Giardia muris. The explanation is included in line 68. This is a review article; therefore, we included the methodology section at the end of the introduction.

Comment 4: “In line 66, the correct term is Giardia intestinalis or duodenalis, but it is written as Lamblia intestinalis.”           

Response: We thank the Reviewer for the suggestion. The term Lamblia intestinalis was used incorrectly. Ultimately, the sentence was removed from the manuscript.

Comment 5: “In line 71, correct the name Lambl; the letter "b" is missing.”

Response: We thank the Reviewer very much for catching this important typo. Professor Lambl's name has been corrected (now line 75).

Comment 6: “In line 77, the concept of an organ is used to refer to the ventral disc; it is better to describe it as a cytoskeletal structure.”

Response: The structure of G. lamblia trophozoite has been described according to the Reviewer's suggestion. The change is visible in the manuscript.

Comment 7: “In line 83, the cyst can contain 2-4 nuclei, not necessarily 2; please correct this.”

Response: We thank the Reviewer for noticing the missing information regarding the number of nuclei contained in G. lamblia cysts. The text has been corrected according to the Reviewer's suggestion.

Comment 8: “The paragraph addressing Giardia assemblages (lines 87-98) contains a significant error: there are 8 Giardia assemblages (A-H) present in different host species. A-B is more common in humans but not exclusive to them, which is why giardiasis is considered a zoonosis, as the authors highlight. There are not two groups of genotypes divided between humans and animals. This needs correction.”

Response: We thank the Reviewer for this comment. The information could have been misleading for the reader and has been corrected according to the Reviewer's remarks (now line 91-100).

G. lamblia exhibits eight unique genotypes (usually designated as assemblage A-H) [5, 6]. Each genotype can cause symptoms of giardiasis, although some are pathogenic. Genotypes A and B are the most common in humans and are found worldwide [21]. Genotypes C-H are less common, and their prevalence depends on the geographical region [5, 8]. The genotypes C-H are characteristic to animal hosts and are primarily associated with infections in animals such as dogs, cats, cows, sheep, pigs, and others [22]. Some of them can also be transmitted to humans, for example, through contact with the feces of infected animals or contaminated water. This results in an overlap of human and animal genotypes, which means that infection with the same species of parasite can occur in both humans and animals.”

Comment 9: “In lines 111-112, it is stated that two trophozoites are released during the excystation of the parasite, which is incorrect. The figure 1 itself indicates the release of an excyzoite (representing a trophozoite that has not completed cell division, which occurs shortly after excystation). This should be corrected.”

Response: The description of the life cycle for the parasite G. lamblia has been revised as follows (now lines 105-115):

“After accidental ingestion, G. lamblia cysts reach the stomach, where they undergo a dynamic transformation called excystation under the influence of hydrochloric acid. This process produces a short-lived stage known as an excyzoite, which then differentiates into four vegetative trophozoites. These trophozoites inhabit the mucous membranes of the small intestine, adhering to them with adhesive apparatuses. They reproduce asexually by longitudinal division, forming new colonies in a process known as multiplication. The massive proliferation of trophozoites reduces the absorptive surface of the intestine and damages the intestinal villi, hindering the absorption of nutrients. Under the action of bile acids, trophozoites undergo encystation, transforming back into cysts that are excreted in the feces. In cases of diarrhea, trophozoites that have not yet encysted are also excreted, contributing to the spread of the infection [25].”

Komentarz 10: „Akapit pomiÄ™dzy wierszami 154 i 160 wymaga odniesienia”. 

Odpowiedź: Odniesienia zostały dodane zgodnie z zastrzeżeniem Recenzenta.

Komentarz 11: „W wierszu 250 wyjaÅ›nij skrót IFA.”

Odpowiedź: OczywiÅ›cie, rozszerzyliÅ›my skrót IFA. Jednak zgodnie z sugestiÄ…, sekcja dotyczÄ…ca diagnostyki zostaÅ‚a ostatecznie usuniÄ™ta z manuskryptu, aby uzyskać krótszy i bardziej przejrzysty tekst.     

Komentarz 12: „Akapit pomiÄ™dzy wierszami 306-321 wymaga podania odniesieÅ„”.

Odpowiedź: Zgodnie z sugestiÄ… ostatecznie usunÄ™liÅ›my z manuskryptu sekcjÄ™ poÅ›wiÄ™conÄ… diagnostyce, aby uzyskać tekst krótszy i bardziej przejrzysty.    

Komentarz 13: „W wierszach 312-313 wspomniano, że hodowla Giardia wymaga skomplikowanego sprzÄ™tu. W rzeczywistoÅ›ci wystarczy prosty inkubator i odpowiednie podÅ‚oże hodowlane”.

Odpowiedź: DziÄ™kujemy Recenzentowi za ten komentarz. Zgodnie z sugestiÄ…, sekcja dotyczÄ…ca diagnostyki zostaÅ‚a ostatecznie usuniÄ™ta z manuskryptu, aby uzyskać krótszy i bardziej przejrzysty tekst.

Komentarz 14: „W sekcji 7, na stronie 9, dla dokÅ‚adnoÅ›ci należy zastÄ…pić „tarcze tarczowe” sÅ‚owem „dysk brzuszny”.

Odpowiedź: DziÄ™kujemy Recenzentowi za ten komentarz. Termin „tarcze dyskoidalne” użyty w sekcji 7, strona 9 (obecnie sekcja 3, strona 4) zostaÅ‚ zmieniony na „dysk brzuszny” zgodnie z sugestiÄ… Recenzenta. Zmiana jest widoczna w manuskrypcie.

Komentarz 15: PeÅ‚na nazwa gatunku powinna być zapisana w tekÅ›cie tylko raz (Giardia lamblia); nastÄ™pnie należy używać formy skróconej (G. lamblia). ProszÄ™ to poprawić w caÅ‚ym tekÅ›cie.

Odpowiedź: W pełni zgadzamy się z tą sugestią. Zmiany zostały wprowadzone w całym tekście.

ChcielibyÅ›my podziÄ™kować Recenzentowi za ogromny wysiÅ‚ek wÅ‚ożony w staranne przejrzenie naszej pracy w dziedzinie parazytologii. Mamy szczerÄ… nadziejÄ™, że nasz artykuÅ‚ wkrótce zostanie opublikowany w specjalnym wydaniu zatytuÅ‚owanym „Parasite Biology and Host-Parasite Interactions: 2nd Edition” czasopisma Molecular Biology. ZaÅ‚Ä…czamy zaktualizowanÄ… wersjÄ™ manuskryptu.

                                                                                                                                                Z poważaniem

                                                                                                                                                Sylwia Klimczak

Reviewer 2 Report

Comments and Suggestions for Authors

Dear authors,

I have read with interest the review entiltled "The Influence of the Protozoan Giardia lamblia on Modulation of the Immune System and Alterations in Host Glucose and Lipid Metabolism" where the authors have found that G. lamblia invasion affects two key phenomena associated with IRrelated metabolic disorders, the inflammatory process and gut dysbiosis, what directly changes metabolic and cytokine profile.

First of all, I think thay usually the reviews comment and compile information, but neve the author found something...unless that you are mixing your results with literature o you are trying a metaanalyses of literature. 

Secondly, everything must be correctly referenced!

And the last, this review is too long and crazy to follow.

Step to step

1. Introduction does not have any refence, whic is unadmissible, and impossible, specially considering the delicate information taht the author gives us.

Anyhow, I would remove this strange introduction and would move the parragraphs to their correct palce through the text

2. The methodology is the last parragraph of the introduction, This is not a scientific paper.

3. Usually a review used the 5 last years. for what did you hHAVE reviewed 15 years??? The local journal, unless that their had been recognozide internationally arre not admissible. The minimun for decent review is  to use Q1 or Q2 journals. By the way, if you use "or" then pubmed searchs one and the other but not both together..... However if you use "and", the pubmed searchs both keyboards in the same paper. Therefore, from my point of wiew I would repeat the search again using "and".

4. the review is too long and wrong structured

5. 3. Giardia lamblia - a single-celled protozoan and 4. Giardiasis - a parasitic disease of the gastrointestinal tract.

Are ok, but I think that it could be better to join (summarizing the thex) under INTRODUCTION. People know the cycle and pathogenesis of Giardia lamblia.  It is unneccesary a class about Giardia. Other option is to name the CDC web page or WHO literature.

6. Methods of Giardia lamblia diagnosis.

It is possible to abreviate andf then to focus on the news perspective such WGS or MALDI-TOF. All methods commented are old well-know methods which trend is dissapear, except PCR or fast-test as IC.

7. Mechanisms of damage to intestinal villi by Giardia lamblia.

This is pathogenesis, then I miss the virulence factors and molecular mechanism of virulence. The summary is ok, but it is lack of knowlwdge and sense. As I said before this is not a class for medical or phramaceutical students. So, please, name the viruelnce factor, expalin correctly the mechanism of virulence and what happen due to that. I do not want a clinical syndromes, I know them.

8. Modulating host inflammatory response

This is the main chapter. It is too long wiyth different information, which must be improved!. Explain each type of response separetly and maybe another chapter talking about CK triger, and after the modulation. So, which type of IGs are delivery, why? Receptor involved. what happen with the natural killers. IL/CK response, why is importan and how works.

Why here the author comments about PG and COX-2 enzyme, why are you mixing this both concepts? Inflamatory and immunology are two different things....

9. Associations between giardiasis and metabolic disorders.

The authors have first forgotten the relation with the microbiota....maybe the alteration or a litle disruption on microbiota could be the triger to lead obesity., too...I will review deeply this chapter. Anyhow, It is too long and a mixed of unsense information.

Thanks

Author Response

Dear Reviewer,

Thank you very much for reviewing our manuscript titled "The Influence of the Protozoan Giardia lamblia on Modulation of the Immune System and Alterations in Host Glucose and Lipid Metabolism" and for providing valuable comments that helped us improve the quality of our work. We have carefully revisited our manuscript and addressed the issues you highlighted. All changes have been made using the "track changes" feature for your convenience. Below, you will find our responses to the comments.

Manuscript ID: IJMS-3076264

Comment 1: “Introduction does not have any refence, whic is unadmissible, and impossible, specially considering the delicate information taht the author gives us.

Anyhow, I would remove this strange introduction and would move the parragraphs to their correct palce through the text”

Response: We thank the Reviewer for this comment. Following the suggestion, we have rewritten the introduction and added references.

Comment 2: “The methodology is the last parragraph of the introduction, This is not a scientific paper.”

Response: We thank the Reviewer for this comment. Since this is a review article, we have added a methodology section at the end of the introduction, as suggested.

Comment 3: “Usually a review used the 5 last years. for what did you hHAVE reviewed 15 years??? The local journal, unless that their had been recognozide internationally arre not admissible. The minimun for decent review is  to use Q1 or Q2 journals. By the way, if you use "or" then pubmed searchs one and the other but not both together..... However if you use "and", the pubmed searchs both keyboards in the same paper. Therefore, from my point of wiew I would repeat the search again using "and".”

Response: We thank the Reviewer for this comment. The phrase "local journal" was used incorrectly. The search for articles has been repeated.  This is a review article on papers published between 2018 and 2024, we have rewritten the introduction and added a methodology section at the end as suggested.

Comment 4: “the review is too long and wrong structured”

Response: Returning to our manuscript after the review period, we understand that it may have been too lengthy and tiring for the reader. Following the Reviewer's suggestion, we have reorganized the chapters and significantly shortened the text. The earlier version included 28 pages of text; the current manuscript has 20 pages. The currently written review article includes: an introduction, 4 main chapters (biology of the parasite, virulence, types of inflammatory response and the relationship of giardiasis to metabolic disorders) and a summary. We hope these revisions have greatly improved the quality of our manuscript and that you find them appropriate.

Comment 5: “3. Giardia lamblia - a single-celled protozoan and 4. Giardiasis - a parasitic disease of the gastrointestinal tract.

Are ok, but I think that it could be better to join (summarizing the thex) under INTRODUCTION. People know the cycle and pathogenesis of Giardia lamblia.  It is unneccesary a class about Giardia. Other option is to name the CDC web page or WHO literature.”

Response: We thank the Reviewer for this comment. As suggested by the Reviewer, the structure of the manuscript has been completely revised.

Comment 6: “Methods of Giardia lamblia diagnosis.

It is possible to abreviate andf then to focus on the news perspective such WGS or MALDI-TOF. All methods commented are old well-know methods which trend is dissapear, except PCR or fast-test as IC.”

Response: We thank the Reviewer for this suggestion. However, the chapter on methods for detecting the G. lamblia parasite has been completely removed from the manuscript. We made this decision to shorten the text, as per the Reviewer's suggestion, and to stay focused on the main message of the work, which is the association between giardiasis, immune response and metabolic disorders.

Comment 7: “Mechanisms of damage to intestinal villi by Giardia lamblia.

This is pathogenesis, then I miss the virulence factors and molecular mechanism of virulence. The summary is ok, but it is lack of knowlwdge and sense. As I said before this is not a class for medical or phramaceutical students. So, please, name the viruelnce factor, expalin correctly the mechanism of virulence and what happen due to that. I do not want a clinical syndromes, I know them.”

Response: The chapter has been revised according to the Reviewer's recommendations. We have provided a detailed explanation of the virulence factors and their mechanisms of action.

Comment 8: “Modulating host inflammatory response 

This is the main chapter. It is too long wiyth different information, which must be improved!. Explain each type of response separetly and maybe another chapter talking about CK triger, and after the modulation. So, which type of IGs are delivery, why? Receptor involved. what happen with the natural killers. IL/CK response, why is importan and how works.

Why here the author comments about PG and COX-2 enzyme, why are you mixing this both concepts? Inflamatory and immunology are two different things....”

Response: We thank the Reviewer for the thorough analysis of this important chapter. In accordance with the Reviewer's instructions, significant revisions have been made: the text has been shortened and divided based on the type of response, and the information regarding PG and COX-2 has been removed (Section 4 - Host inflammatory response and its modulation; line 225-364). 

Comment 9: “Associations between giardiasis and metabolic disorders.

The authors have first forgotten the relation with the microbiota....maybe the alteration or a litle disruption on microbiota could be the triger to lead obesity., too...I will review deeply this chapter. Anyhow, It is too long and a mixed of unsense information.”

Response: We thank the Reviewer for this comment. The reference to the relationship with the microbiota has been revised, and the chapter has been shortened to include only the most important information.

We would like to thank the Reviewer very much for their tremendous effort in carefully reviewing our work in the field of parasitology. We sincerely hope that our article will soon be published in the journal Molecular Biology in the special issue titled "Parasite Biology and Host-Parasite Interactions: 2nd Edition." We have attached an updated version of the manuscript.

                                                                                                                                               Yours sincerely

                                                                                                                                               Sylwia Klimczak

Round 2

Reviewer 1 Report

Comments and Suggestions for Authors

There is one mistake into the scheme of Giardia differentiation. After excystation 2 (two) trophozoites are released, not four (4) as showed in the figure 1 and stated in the figure legend. Please correct it.

Author Response

Dear Reviewer,

I hope this message finds you well. Thank you for your feedback on our manuscript titled "The Influence of the Protozoan Giardia lamblia on Modulation of the Immune System and Alterations in Host Glucose and Lipid Metabolism" (ID: ijms-3076264).

Comment: “There is one mistake into the scheme of Giardia differentiation. After excystation 2 (two) trophozoites are released, not four (4) as showed in the figure 1 and stated in the figure legend. Please correct it.”

Response: We thank the Reviewer for this comment. We updated Figure 1 and the corresponding description to accurately reflect that two trophozoites are released after excystation (the changes are visible in line 113). Thank you for highlighting this mistake and helping us improve our manuscript.

We also appreciate your willingness to review the text again and allow us to make corrections. We hope that the revised version of the manuscript meets Reviewer’s expectations, and that the publication will be accepted soon.

Sincerely,

Sylwia Klimczak

Reviewer 2 Report

Comments and Suggestions for Authors

My suggestion does not have been reach

Author Response

Dear Reviewer,

I hope this message finds you well. Thank you for your feedback on our manuscript titled "The Influence of the Protozoan Giardia lamblia on Modulation of the Immune System and Alterations in Host Glucose and Lipid Metabolism" (ID: ijms-3076264).

Comment: “My suggestion does not have been reach”

Response: Dear Reviewer, we have reviewed the manuscript and made changes according to your suggestions. The most noticeable change compared to the previously submitted text concerns Chapter 4, which discusses the inflammatory response in detail.

We have incorporated a more concise and focused description of the immune response to G. lamblia, emphasizing the key points. Specifically, we described how G. lamblia attaches to enterocytes, causing mechanical damage and initiating an immune response. This response involves the activation of various immune cells (macrophages, dendritic cells, neutrophils, NK cells) via PRR receptors, particularly TLR2 and TLR4, leading to the release of pro-inflammatory cytokines (IL-12, TNF-α, IL-6) and activation of the complement system.

In the adaptive response, we highlighted the roles of CD4+ T cells, including Th17 and Th1 cells, which produce IL-17 and IFN-γ, respectively, to recruit neutrophils and activate macrophages. The production of specific antibodies (IgA, IgM, IgG, IgE) against the parasite was also detailed, explaining their roles in neutralizing and facilitating the removal of the parasite.

We also included information on how G. lamblia modulates the host immune response by changing its surface antigens, degrading cytokines and chemokines, and impacting the gut microbiome. This modulation helps the parasite evade the immune system and persist in the host.

Other minor changes are also visible in the manuscript. We hope that your expectations have been met; if not, we kindly ask you to indicate the areas that need improvement.

Thank you for your time and consideration.

Sincerely,

Sylwia Klimczak